# NIMO: A Natural Product-Inspired Molecular Generative Model Based on Conditional Transformer

**DOI:** 10.3390/molecules29081867

**Published:** 2024-04-19

**Authors:** Xiaojuan Shen, Tao Zeng, Nianhang Chen, Jiabo Li, Ruibo Wu

**Affiliations:** 1School of Pharmaceutical Sciences, Sun Yat-sen University, Guangzhou 510006, China; shenxj9@mail2.sysu.edu.cn (X.S.); zengt28@mail2.sysu.edu.cn (T.Z.); nhchen95@gmail.com (N.C.); 2ChemXAI Inc., 53 Barry Lane, Syosset, NY 11791, USA

**Keywords:** natural products, molecular generation, deep learning, fragmentation, transformer

## Abstract

Natural products (NPs) have diverse biological activity and significant medicinal value. The structural diversity of NPs is the mainstay of drug discovery. Expanding the chemical space of NPs is an urgent need. Inspired by the concept of fragment-assembled pseudo-natural products, we developed a computational tool called NIMO, which is based on the transformer neural network model. NIMO employs two tailor-made motif extraction methods to map a molecular graph into a semantic motif sequence. All these generated motif sequences are used to train our molecular generative models. Various NIMO models were trained under different task scenarios by recognizing syntactic patterns and structure–property relationships. We further explored the performance of NIMO in structure-guided, activity-oriented, and pocket-based molecule generation tasks. Our results show that NIMO had excellent performance for molecule generation from scratch and structure optimization from a scaffold.

## 1. Introduction

Natural products (NPs) are derived from evolutionary selection over millions of years to bind to biological macromolecules and therefore possess important biological activity and pharmaceutical value [1]. With the rapid development of pharmacology and synthesis, more and more natural products are coming to our attention as an important source of new bioactive compounds with novel molecular scaffolds [2]. According to a comprehensive study, 6% of all small-molecule drugs approved between 1981 and 2014 are unaltered NPs, 26% are NP derivatives, and 32% are NP mimetics and/or contain an NP pharmacophore [3]. Judging by the average number of natural product-derived fragments (NPFs) in approved drugs since 1939, pharmaceutical drug discovery programs continue to benefit from the use of NPFs [4,5]. As far as we know, NPs have high diversity and structural complexity, such as a high fraction of sp^3^ carbon atoms, stereogenic centers, and diverse ring systems, which make them a largely unexplored chemical space and able to be widely incorporated into the pipelines of drug design on a large scale [6].

Inspired by pre-validated NP repositories in nature, e.g., biology-oriented synthesis [7,8] (BIOS) and pseudo-natural product (pseudo-NP) strategy [9,10] (Figure 1), many novel biologically relevant compounds are designed and synthesized. For BIOS, a conserved core scaffold is identified during the lead identification phase and often kept throughout the rest of the compound collection design. Scaffold synthesis and decoration following BIOS could yield new compounds. On the other hand, for pseudo-NPs, the biological relevance of NPs merges with the rapid accessibility through fragmentation and reassembly, going beyond existing NP scaffolds into unexplored chemical space to overcome the limitations of BIOS [11,12]. Overall, dynamic combinatorial chemistry [13] plays an important role in natural product research.

Computationally, many de novo molecular generative models are aimed at generating compound structures with desired physicochemical and bioactivity properties or even multi-objective optimization [14,15,16]. In terms of granularity, SMILES (simplified molecular-input line-entry system) strings are often adopted as a molecular representation due to their simplicity. For instance, MCMG [17] is a multi-constraint molecular generation approach based on a transformer [18] for de novo drug design. QCMG [19] is a quasi-biogenic molecule generator with recurrent neural networks. However, SMILES-based models often undergo substantial changes during sequential extensions [20]. For example, two molecules with similar chemical structures may be encoded into significantly different SMILES strings. The impact of this characteristic on the structures of polycyclic complex natural products has not been adequately assessed. Some latecomers of graph generation schemes, such as JT-VAE [21], which generates graphs in a motif-by-motif manner rather than node-by-node, are employed to obviate chemically invalid intermediates. Such generators belong to the same fragment-based model as FBMG [22] with respect to the granularity of their applied molecular representation. In this context, there are still relatively few generation models focused on natural products. Given that new chemical entities are typically derived from structural modification of active natural products obtained through screening techniques [23,24,25], there is a substantial demand for multi-objective structural optimization in natural product-derived models, such as the derivation of the scaffold, in addition to de novo generation [26]. In particular, fragment-based paradigms are thought to be suitable for this real scenario [27,28,29]. The scaffold-based models typically support scaffold as the initial seed of the generative procedure [30,31]. Last but not least, natural products often feature biologically relevant molecular scaffolds and pharmacophore patterns [4,32]. However, the current molecular generators ignore this critical transfer of relevant structural features to NP-inspired compound libraries.

To our knowledge, previous generative models appear to be confined in the face of the following critical challenges towards natural products: (1) manipulability for complex natural product structures including stereo information, (2) multi-objective structural optimization, and (3) inheritance of biological relevance from natural products. Thus, flexible generative models are needed to complement routine design strategies in real scenarios. In this work, we develop a natural product-inspired molecular generative model (called NIMO for short) based on transformer architecture, in which the rich semantic information among motifs from the aforementioned BIOS and pseudo-NP strategies is learned, and construct the NIMO-S and NIMO-M models, respectively, for generating natural product-like molecular structures that comply with the expected criteria. Specifically, NIMO-M is a generic model for molecular generation with multi-constraint and novel motifs, while NIMO-S is a scaffold-based model for lead optimization that specifies a central scaffold. Furthermore, we thoroughly investigated NIMO in the multi-objective molecular generation tasks and show that NIMO excels in several classical methods in three practical tasks, covering structure-guided, activity-oriented, and pocket-based molecule generation. 

## 2. Results and Discussion

### 2.1. Model Evaluation

We first trained and sampled 5000 molecules under evaluation setting 1 (see Section 4.3). As shown in Figure 2, the reconstructed chemical spaces of NIMO-S and NIMO-M exhibited a spatial contraction toward the desired properties (QED, logP, and SAS) in contrast to the native chemical space of the NPs. It should be noted that the Mw (molecular weight) distribution was optimized due to the correlation with QED. The statistics showed similar and slightly concentrated molecular property distributions for HBD, HBA, and RB. To assess whether the method can capture the intrinsic structure features of natural products, NP-likeness [33] was introduced as a measure of similarity to the NP molecules, and it showed that both models could generate molecules with more preferred features of NP-like compounds than the synthetic molecules in ZINC [34]. 

We reported the benchmark studies of 5000 generated molecules under evaluation setting 2 (see Section 4.4). Here, we used SMILES-based models (MCMG and QBMG) and a fragment-based model (FBMG) as the baselines, and the conditional metrics and MOSES [35] metrics were utilized as comprehensive evaluation benchmarks. Table 1 illustrates that all models except FBMG performed pretty well for the validity rate (above 90%). In particular, NIMO not only had significantly higher fragmentation efficiency than FBMG but also yielded a smaller motif size and lower motif weight (Appendix A). Because FBMG had difficulties in proposing valid molecules, in addition to its inability to handle stereochemical information and multi-constraint generation, it was not considered in later analysis. On the other side, the validity of the molecules generated by NIMO-M dropped down to 75.12% when the motif information was removed from the training set. This verifies that additional motif information was conducive to model training, thus guaranteeing model reliability and improving training efficiency. Details of the ablation experiment are available in Appendix A. All models scored high on the uniqueness indicator. 

In terms of novelty, NIMO-S offered the best performance, while NIMO-M underwent the sharpest drop, to 61.0%. The relatively underperforming MCMG achieved 65.7%. According to MOSES metrics, NIMO-M and MCMG consistently performed well in terms of FCD metrics, which were related to chemical and biological properties. Compared to the SNN metrics, the structures generated by NIMO-S were the furthest from the manifold of the training set. The Frag and Scaf metrics compared molecular similarities at the substructure level. Note that the metric calculation method applied the Bemis–Murcko scaffolds, which partially overlapped with our motif extraction method, so it was comprehensible for the enhancement of Frag metrics of NIMO-S. All models showed roughly the same IntDiv of the generated molecules, indicating the diversity of the generated molecules. NIMO-M achieved the most impressive performance in terms of synthetic accessibility (SAS), demonstrating the practical applicability of the model for molecule generation.

Overall, NIMO achieved a major breakthrough as a fragment-based model compared to the FBMG and is in no way inferior to the state-of-the-art SMILES-based model. Indeed, novelty remains a future endeavor in the field of molecular generation. Next, three practical molecular generation tasks, including structure-oriented (terpenoids), bioactivity-oriented (antimalarials), and target-driven (antibacterial) tasks, were performed to demonstrate the applicability of our model. 

### 2.2. Terpenoid Generation

In order to evaluate whether the model can generate molecules targeting specific structure regions, we specifically designed the generation task of anchoring complex polycyclic terpenoids, which is the biggest class of NPs. As shown in Table 2, four models were trained based on the TeroKIT database. Herein, NIMO-S’ added ring separation and ring recombination (edge fusion) functions to NIMO-S to compensate for the lack of scaffold diversity. The additional step was helpful for reducing the size and weight of the motifs, as shown in Appendix A. Subsequently, all generated molecules were judged in terms of whether the structure belonged to terpenoids by NPClassifier [36]. As summarized in Table 2, NIMO-S’ had the best performance (95.4%) for effectively constructing terpenoid compounds, followed by NIMO-S, QBMG, and QCMG. This indicates that our NIMO models can generate more norm-compliant molecules with the constraints of the established structure rules. 

The functional groups (FGs) and ring systems (RSs) were then identified for the 5000 generated molecules and the TeroKit dataset [37,38]. As shown in Table 2, NIMO-S and NIMO-S’ exhibited good coverage of scaffolds present in the training set, according to “coverage” and “recovery”. Obviously, our scaffold-based NIMO model can maximally reproduce the substructural features of the original training set. In addition, a growing body of evidence supports the effectiveness of retaining specific substructures (e.g., core scaffolds) or general structural features (e.g., RSs and FGs) for inheriting the biological relevance of natural products [39,40]. This is of greater importance for the structural modification of proven scaffolds in drug screening. Therefore, we further developed a scaffold-based scenario for a more elaborative evaluation. 

Two motifs were chosen as core seeds and utilized for molecular generation (Figure 3). It was found that NIMO-S could reproduce the same modifications at conserved sites of the core scaffold. Moreover, due to the correct definition of the extension sites, it offered diverse modifications for the scaffold with different functional groups. For example, the same functional group modifications, such as methyl and hydroxyl groups, were present in the generated molecules at some derivation sites (e.g., C-4) compared to the real molecules seen from scaffold 1. More importantly, NIMO-S decorated various substructures at extension sites to generate diverse derivatives, such as the long side chain at C-11 of scaffold 1 (blue) and C-21/16 of scaffold 2 (green). More structural analysis of the generated molecules is provided in Appendix A, and Appendix A depicts that NIMO was also able to reconstruct a similar chemical space of terpenoids. 

### 2.3. Antimalarial Activity-Oriented Molecular Generation

As NPs serve as the major source of lead compounds against malaria [41,42], the motif-based NIMO model was used to discover potent new antimalarials. Large-scale predictions of potential antimalarial compounds were made on MAIP [43]. Valid molecules were sampled from models under evaluation setting 3 (see Section 4.5). As a result, the histogram distribution of the predicted scores from MCMG roughly followed a normal distribution, like the training data, whereas those from NIMO-M appeared unsmoothed and discontinuous (see details in Appendix A). The antimalarial activity prediction of molecules generated by NIMO-M and MCMG are summarized in the left four columns of Table 3. NIMO-M excelled in two of the three enrichment factor metrics and outperformed MCMG, with an overall activity rate of 55.9%, approximately 46% higher than the training dataset. 

The first quartile, indicating the novel molecules generated by two models, was quantitatively close, as shown in Figure 4a. NIMO showed dense enrichment at high similarity around the third quartile. Four high-frequency motifs in the top 10% of active molecules generated by NIMO-M are listed in Figure 4b. Molecules containing high-frequency motifs yielded a more dominant predicted score, as shown in Figure 4c. This reflects that the NIMO models were capable of sampling active motifs, which made up a large part of the total sampling volume. In particular, Figure 4d shows that Motif2 had more potential for exploring antimalarial activity. Thus, Motif2 was seeded into the trained NIMO-M for resampling, which was named NIMO-M’. As we expected, NIMO-M’ achieved a boost in the enrichment ability toward anti-malarial activity. As shown in the rightmost column of Table 3, three enrichment factors were significantly increased, with the overall activity ratio raised by approximately 75.5% over the training data. The result also shows that the predicted activity was more susceptible to fragments than tokens tokenized by SMILES. To facilitate data analysis, we also visualized the chemical spatial distribution of the training set and generated molecules using TMAP [44], as shown in Appendix A. If we regard the location of the NIMO-M’-generated molecules as the highly active region (orange dots), then we can observe that the NIMO-M-generated molecules were clustered nearby, resulting in a high molecular density. This illustrates that NIMO-M exhibited a structural preference over the active region formed by the dominant motifs, which is advantageous for realistic molecular generation practices. 

### 2.4. Pocket-Based Molecular Generation

NIMO showed excellent enrichment ability in the above specific activity-oriented task. Next, we spotlighted the fragment-derived methods and strategies for the effectiveness of virtual library development [45,46]. We proposed a general approach to design antibacterial discovery libraries. Briefly, we collected an antibacterial dataset against experiment-relevant Gram-positive and Gram-negative bacteria, covering fifteen common bacterial species. Then, a phenotypic antimicrobial model based on a kNN classifier against the bacteria panel was constructed by molecular fingerprinting analysis. Then, the classifier was used to further predict targeted bacterial species for molecules generated by NIMO-M. Finally, we obtained a target annotated intelligent library. See Appendix A for detailed steps.

Here, the Lactobacillus-targeted compounds from the above antibacterial library were selected for further analysis. The compounds falling under the targets (2HMG, 1BO7) were docked into the associated protein pockets, as shown in Table 4. According to docking results by MOE [47], among the 5000 molecules generated for 2HMG, 82 candidates were predicted by molecular fingerprinting analysis, and the docking scores of 26 molecules were lower than those of the native ligand (the lower score the better). For the top 1000 molecules, 10 of the 15 candidates had a dominant docking score. There was also a significant proportion of compounds for 1B07. Moreover, compounds with RMSD values of less than 2 Å to native ligands indicate that the original binding poses could be well recovered by NIMO.

Figure 5 showed four high-quality binding poses of compounds against Lactobacillus selected from the above virtual antibacterial library. There were favorable MOE docking poses after overlay with native ligands in three protein pockets. Besides a high 3D shape similarity, they had a better docking score compared to the co-crystal ligand, indicating a positive binding affinity with the pocket. Meanwhile, three indicators (QED, SAS, SI) also showed that NIMO could deliver chemically reasonable compounds. On the other hand, compounds 3 and 4 appeared to have the same topological structures of 2,4-diaminopyrimidine rings (in a circle with dashed lines) with native ligands [48,49]. This suggests that NIMO can both reproduce the key pharmacophore features of the active ligand and capture more attractive fragments from the training set.

### 2.5. Discussion

NIMO is a fragment-based generator capable of handling stereochemical information from natural products. In the case of NIMO-M, the attachment points that tagged the cleaved bonds retained chiral information in the initial fragments after fragment extraction. Moreover, our model also accounted for constraint structural optimization and allowed derivative compounds to be formed starting from a specified substructure, which was extremely useful in practice. In addition to the above innovations to address basic challenges, the most notable differences were in the performance for NIMO in comparison to other methods. (1) NIMO can generate more norm-compliant structural categories under the intended structural disciplines. For example, our model generated terpenoid structures with a success rates of 95.4% (NIMO-S’) and 91.9% (NIMO-S), which outperformed the baseline models in the structure-based generation task (Table 2). (2) The molecules generated by NIMO can inherit the biological relevance in a friendly way by maximizing the reproduction of substructures found in natural products. For example, our model reproduced ring systems and functional groups that were pre-existing in the training dataset, surpassing baseline models in terms of coverage and recovery metrics (Table 2). (3) NIMO can detect potentially privileged motifs that contribute to activity and enrich more active molecules as a result (Figure 4). Meanwhile, NIMO showed a strong structural preference for highly active regions rather than a uniform distribution (Appendix A). (4) In terms of a granularly fragment-based algorithm, the high efficiency of fragment extraction brings smaller motifs and consequently increases the molecular diversity. This was confirmed in the comparison result of motif extraction among fragment-based models (Appendix A). For instance, the mean weight of the motifs produced by our model was 218.5 g/mol (NIMO-M) and 238.2 g/mol (NIMO-S), significantly lower than that of the same fragment-based model, FBMG, which stood at 407.2 g/mol. On the other side, the high reconstruction accuracy (99.9%) warrants that only the correct motif sequences were fed into the model; thus, it circumvented the puzzles of where to attach the new fragment and which chemical bond to choose, as required by conventional fragment-based models. Nevertheless, the future development of NIMO still comes up against a few open questions. NIMO can mimic fragment rearrangement, ring separation, and ring combination (edge fusion), but some other more complicated design strategies, such as opening/closing ring and bridged ring, were not realized to generate pseudo-NPs in the current work [11,50,51].

## 3. Methods

### 3.1. Data Preparation

In this work, all available datasets were collected from public domains, including COCONUT [52], TeroKIT [53,54], ChEMBL [55], and the study of Andreas Verras et al. [56], as detailed in Section 4.1. The data were filtered according to molecule standardization for consistency. The complete procedure consisted of desalination, charge neutralization, removal of glycosylation, and checking of molecular validity. Also, duplicates were removed. As a result, all natural products with stereochemistry were collected in the form of canonical SMILES strings. The filter was followed by calculating each molecular entry with molecular descriptors. They were used as constraints for the training model and as metrics for the model evaluation.

### 3.2. Motif Sequence Generation

We utilized two tailor-made methods to generate motif sequences for NPs, applied to the motif-based model (NIMO-M) and the scaffold-based model (NIMO-S), respectively. First, we defined a motif $S_{i}\;$ as a subgraph of molecule G. Second, we decomposed molecule G into fragments by breaking bonds specifically by the fragmentation rules. Many rules were included, but not limited to the BRICS [57] and Murcko [58] fragmentation methods (see Section 4.2). The generated fragments contained some dummy atoms with their original bond IDs, allowing the original connection to be memorized. See Appendix A for details of the fragmentation protocol. Next, the generated fragment sequence was canonicalized according to the dummy atom IDs of the initial fragments. The sequence order was generated in such a way that the original molecule could be reconstructed without using the dummy atom IDs. We denoted each canonicalized fragment with rich semantic information as a motif $S_{i}$, and the motif served as the basic unit for training the model. The procedure for canonicalizing fragmented sequences is provided in Appendix A, and two cases of motif sequence generation are presented in Appendix A.

### 3.3. Molecular Reconstruction Verification

Molecular reconstruction verification was applied after the canonical motif sequences were generated. The verification was used to filter out a small number of invalid sequences, as illustrated in Appendix A. An example of molecular reconstruction is outlined in Appendix A. A rigorous examination of all motif sequences ensured a molecular reconstruction accuracy of 99.9%, regardless of chirality differences. In contrast, the HierVAE-decoder [21] also utilized motifs as building blocks for generation, though it only reached an accuracy of around 80%. This indicates that the reserved dummy atoms in the motif allowed us to omit the process of attachment prediction and reduce the loss. This significant boost gave us great confidence in the reliability and interpretability of the fragment-based model.

### 3.4. Model Architecture

As shown in Figure 6, the core of NIMO used a conditional transformer architecture to generate NP-derived molecules with desirable properties. First, each pre-processed input sequence, including constraints, motif information, and motif sequence, was viewed as a sentence and a vocabulary was constructed. “Motif info” represents the number of attachment points automatically extracted from the motif sequence in advance. Second, the sentence was fed into input embedding, followed by the addition of positional encoding. Here, the standard sinusoidal positional encoding allowed the transformer to preserve the relative position of words in a sentence. The core architecture consisted of multiple decoder stacks. Each decoder layer had a multi-head self-attention sub-layer and a position-wise feedforward network (FFN) sub-layer. The masked multi-head self-attention layer ensured that the prediction of the current position relied only on the sequence embedding information prior to that position. The self-attention layer applied scaled dot-product attention functions and facilitated the model to capture information from different subspaces at different positions. The formula of the attentional mechanism can be described according to the following equation:
(1)$$Attention(Q,K,V)=softmax(\frac{{QK}^{T}}{\sqrt{d_{k}}})V$$

This formula required the introduction of the query (*Q*), a key (*K*), and a scaling factor *d_k_*. Then, a softmax function was used to obtain the weights of the values (*V*). The FFN adopted two layers of fully connected layers. Next, ReLU was an activation function followed by a layer normalization procedure. Then, a residual connection was applied to ease the gradient disappearance and allow for a deeper network. The decoder outputs yielded a probability distribution over all latent semantic rules for each time step. The input sequence was expressed as $X=x_{1}$, …, $x_{k}$. Since the model inputs contained desirable properties (as a constraint condition c), the model was trained to minimize the following negative log-likelihood:(2)$$L\left(x\right)=-\sum_{i=1}^{k}log{\;p\;(x}_{i}|{x_{0}}_{\;},\ldots ,\;x_{i-1},c)$$

During model sampling, linear and softmax layers produced an output probability for the next word according to a learnt conditional probability distribution:(3)$$x_{i}\sim p\left(x_{0},\ldots ,\;x_{i-1},\;c\right)$$

The sampling problem was defined as the search for the most probable hypothesis y* according to a trained model and a set of constraints c. If V was the search space formed by vocabulary combinations, $y^{*}$ was calculated by the following equation:(4)$$y^{*}=argmax\;p\left(y|x,c\right)\;y\in V$$

The decoder recursively generated the subsequent samples by adopting a beam search algorithm [59]. The beam search kept the K locally highest probability candidates at each time step t, where the hyperparameter K was referred to as the beam width. The recursion was performed until all sampling sequences ended in the character “EOS” or the predefined maximum time step T was reached. The maximum search space of this algorithm in one generation process was related to the spatial complexity O(TKV). More details on the sensitivity analysis of parameter K can be found in Appendix A. Herein, we modulated the probability distribution in some scenarios by avoiding the occurrence of unreasonable fragments and preventing the beam from going in a repetitive direction. Next, top N hypotheses $y^{*}$ were selected from the searched set according to scoring accumulated probabilities. Finally, a molecular reconstruction algorithm transformed the N sequences to the final molecular structure, which was a reverse procedure of the molecular fragmentation and canonicalization.

## 4. Experiment Configuration

### 4.1. Datasets

#### 4.1.1. COCONUT

The natural product structures in the training dataset were downloaded from the collection of open natural products (COCONUT, https://coconut.naturalproducts.net/ (accessed on 1 May 2023)) in absolute SMILES (includes stereochemical information) format, which initially contained about 744,986 unique canonical SMILESs.

#### 4.1.2. TeroKIT

The terpene dataset TeroKIT was obtained from our group’s previous study. About 173,914 annotated terpenoids were collected. More details can be accessed at (accessed on 1 May 2023).

#### 4.1.3. Anti-Malarial Experimental Activity Data Set

The anti-malarial experimental activity dataset was MMV-St. Jude, which was obtained from the study reported by Andreas Verraset al. It contains 2507 positive compounds.

#### 4.1.4. Antibacterial Dataset

All antibacterial compounds against common Gram-positive and Gram-negative bacteria were retrieved from the ChEMBL dataset (255,788).

### 4.2. Fragment Extraction

Given a compound library, we used two special fragmentation methods to transform the molecular graph into a sequence of fragments, which were applied to NIMO-M and NIMO-S. First, we defined a motif $S_{i}=\left(V_{i},E_{i}\right)$ as a subgraph of molecule G, where $V_{i}$ is the set of atoms (vertices) and $E_{i}$ is the set of bonds (edges). To extract motifs, we decomposed molecule G into fragments by breaking bonds specified by the following rules. In NIMO-M, (1) find all the single bonds (μ, ν) ∈ E, where u is in a ring, and ν is in an off-ring or is in another ring. Bonds (μ, ν) are undirected. (2) Find all the bonds that meet BRICS [57]. In NIMO-S, (1) find all the bonds between the Murcko scaffold and the side chains. (2) Find a bond (u, v) in Murcko [58] scaffold that represents a shared edge in a fused ring. Meanwhile, the bond (μ, v) divided the Murcko scaffold S_1_ future into two subgraphs (S_2_, S_3_). The fragment extraction resulted in initial fragments containing dummy atoms. Therefore, we obtained motifs such as “C1CC[*][*]C1” and “[*]1 = [*]C = CC1”, where the atom types u and v were further replaced by dummy atom [*].

### 4.3. Evaluation Setting 1

The training data were from the COCONUT dataset. A total of 5000 molecules were sampled from the multi-constraint models. Specifically speaking, three of these molecular features were selected as constraints to train the models in our work. The QED, logP, and SAS were expressed as scalars. Each molecule was labeled with different attributes based on a customized threshold value, such as “good logP”. These labels were applied to train the biased model as constraint codes. Model training and optimization of hyperparameters are provided in Appendix A. Finally, we plotted the distributions by the statistics and analysis of partial descriptors.

### 4.4. Evaluation Setting 2

FBMG was a fragment-based generative model as well. QBMG was a natural product-focused SMILES-based generative model. MCMG was one of the most advanced SMILES-based generative models and was also used for contrast. It should be pointed out that MCMG in this article specifically refers to MCMGM, where distilled molecules (DM) were taken as the knowledge distillation method. NIMO was trained just like evaluation setting 1.

### 4.5. Evaluation Setting 3

First, predictions of potential malaria-inhibiting compounds from the COCONUT dataset were made on MAIP. The ultimate predicted output of MAIP is a model score. Here, we defined 44.36 as a score threshold. This meant that 10% of natural products with a model score > 44.36 were labeled as anti-malaria active data, while the remaining 90% were labeled as inactive. Moreover, a portion of the anti-malarial experimental activity dataset (2507) was coupled with labeled compounds to train the models. Finally, 5000 valid molecules were sampled from models conditioned on a given activity constraint. Only MCMG was allowed as the baseline model to perform activity-constraint molecular generation.

### 4.6. Baseline Models

#### 4.6.1. MCMG

We downloaded the code from the official repository https://github.com/jkwang93/MCMG (accessed on 1 July 2023). The MCMG as a SMILES-based model was unable to handle stereo information. There were 316,864 unique “flat” (with no stereochemistry) NPs in the training set after the elimination of stereo information and de-duplication. Additionally, MCMG was slightly modified to carry out its training process with the same constraints as ours (QED, logP, SAS). Then, the model was trained according to the process described in the original study.

#### 4.6.2. QBMG

We downloaded the code from the official repository https://github.com/SYSU-RCDD/QBMG (accessed on 1 July 2023). QBMG was able to handle stereo information as a quasi-biogenic molecule generator. QBMG was trained without constraints in order to avoid major human intervention.

#### 4.6.3. FBMG

We downloaded the code from the official repository https://github.com/marcopodda/fragment-based-dgm (accessed on 1 July 2023). FBMG was trained without constraints in order to avoid major human intervention and maintain the original function. The model cannot generate molecules with stereo information. Default parameters were used to train the model.

### 4.7. NP-Likeness Score

The NP-likeness score was calculated using RDKit-based implementation of the method described in the original article, which can be found in the repository https://github.com/rdkit/rdkit/tree/master/Contrib/NP_Score (accessed on 1 March 2023).

### 4.8. NPClassifier

The NPClassifier is a deep learning-based automated structural classification of NPs. Herein, the final statistic in Table 2 depends on the terpenoids classified by the NPClassifier. The detailed implementation can be found by visiting this file: https://pubs.acs.org/doi/suppl/10.1021/acs.jnatprod.1c00399/suppl_file/np1c00399_si_003.pdf (accessed on 1 July 2023).

### 4.9. MAIP

The malaria inhibitor prediction (MAIP) is accessible through https://www.ebi.ac.uk/chembl/maip/ (accessed on 1 July 2023). When using the web service to predict blood-stage malaria inhibitors, MAIP returns a predicted model score. A higher score means greater enrichment.

## 5. Conclusions

In this work, we proposed a new design strategy (named NIMO) for molecule generation to efficiently explore the vast chemical space of natural products. NIMO is helpful for discovering bioactive NP-like compounds and structural modification of NPs. Two sets of motif extraction methods were used to fragment molecule structures and derive motifs in semantically meaningful sequences. A constrained transformer framework was developed to capture rich semantic information and implicit linking rules. As a result, NIMO demonstrated superior performance across three typical applications (structure-guided, activity-oriented, and pocket-based). Although there is still room for further improvements, we believe that NIMO could provide a general computational framework for fragment-to-lead design to accelerate the construction of high-quality pseudo-natural product libraries. This approach can be applied to various scenarios, such as multi-objective structural optimization, scaffold-based lead optimization, and activity-oriented enrichment based on dominant fragments, thereby facilitating drug discovery for natural products.

## Figures and Tables

**Figure 1 molecules-29-01867-f001:**
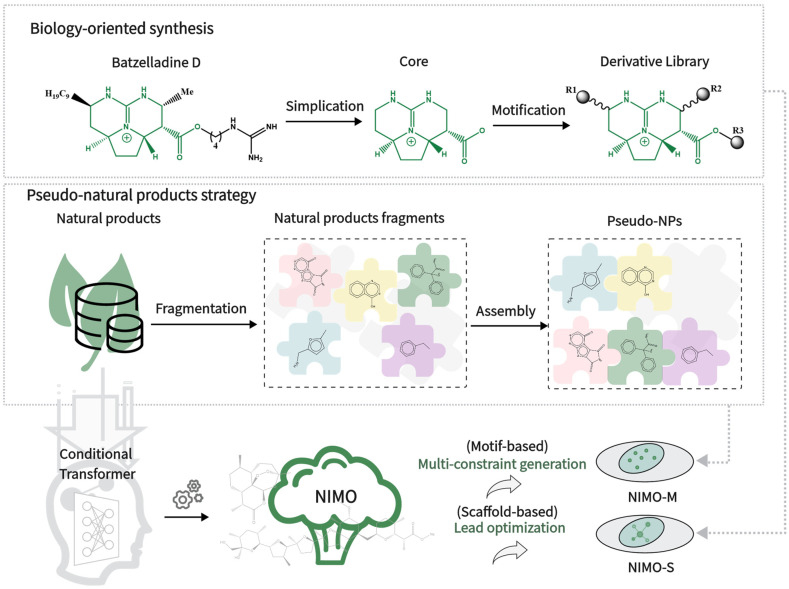
The two bio-inspired ideas integrated into our developed NIMO.

**Figure 2 molecules-29-01867-f002:**
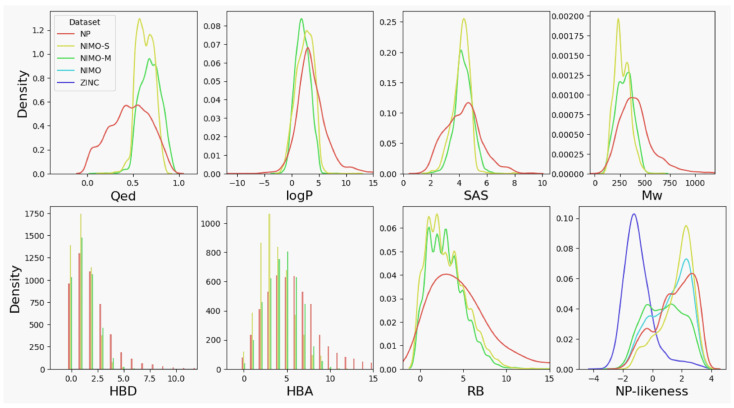
Property distributions of the molecules generated by NIMO. NP (red) refers to the train set, while the ZINC (purple) dataset is identified as synthetic molecules. NIMO (blue) is from the integration of molecules generated by NIMO-M (green) and NIMO-S (yellow). QED is “quantitative estimate of drug likeness”; logP indicates “octanol/water partition coefficient”; SAS indicates “synthetic accessibility score”; Mw indicates “molecular weight”; HBD indicates “hydrogen bond donor”; HBA indicates “hydrogen bond acceptor”; RB indicates “rotatable bond”.

**Figure 3 molecules-29-01867-f003:**
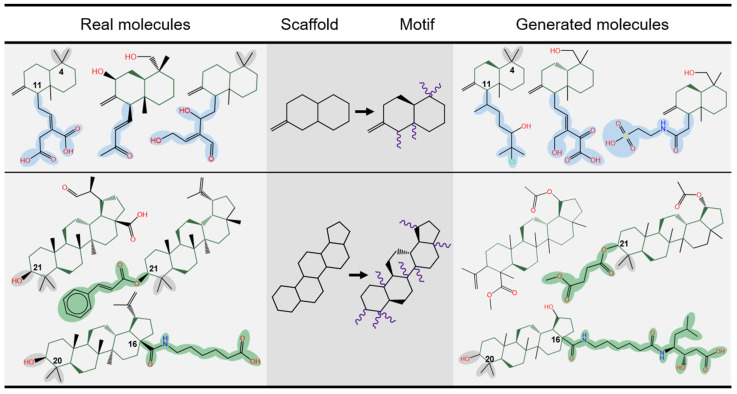
The typical cases of derivatives generated by NIMO-S in a scaffold-based scenario. The middle columns are motifs corresponding to the two randomly selected scaffolds from the terpenoid training set. The left and right columns are real molecules containing the structure of scaffolds and molecules generated by NIMO-S, respectively. The extension sites are highlighted in purple.

**Figure 4 molecules-29-01867-f004:**
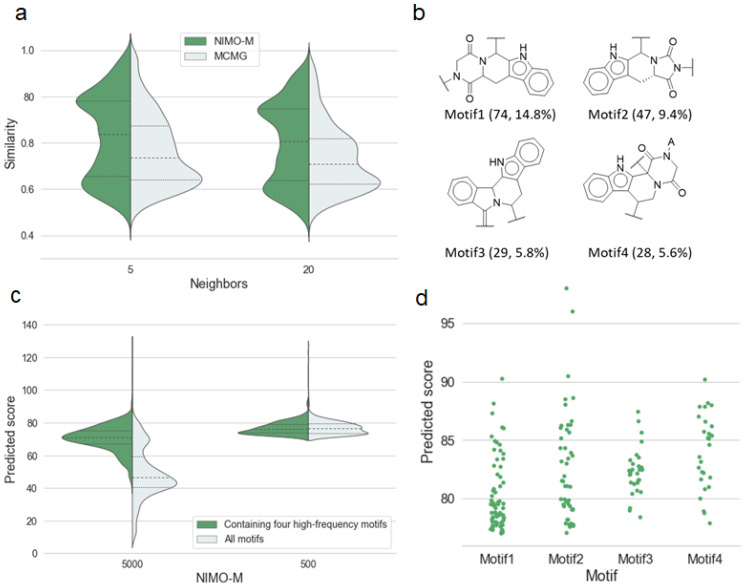
(**a**) Distribution of the average Tanimoto similarity between fingerprints of molecules generated by models (NIMO-M, MCMG) and the nearest neighbor molecules from the training set. (**b**) Four high-frequency motifs in the top 500 molecules generated by NIMO-M ranked by anti-malarial activity score, along with the number and percentage of molecules categorized by the motifs in brackets. (**c**) Violin plots of predicted scores for molecules generated by NIMO-M. The left represents the 5000 molecules, and the right represents the top 500 molecules scored by predicted scores. (**d**) Scatter plots of predicted scores of molecules categorized by the above four motifs.

**Figure 5 molecules-29-01867-f005:**
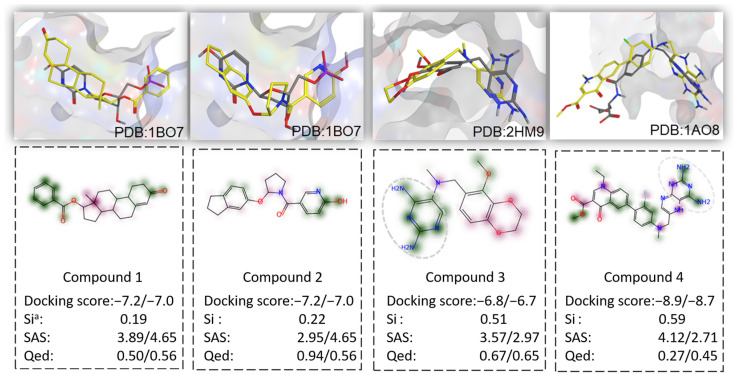
Examples of reasonable compounds generated by NIMO-M. ^a^ SI represents similarity index. The data behind the “/” is the value for native ligands.

**Figure 6 molecules-29-01867-f006:**
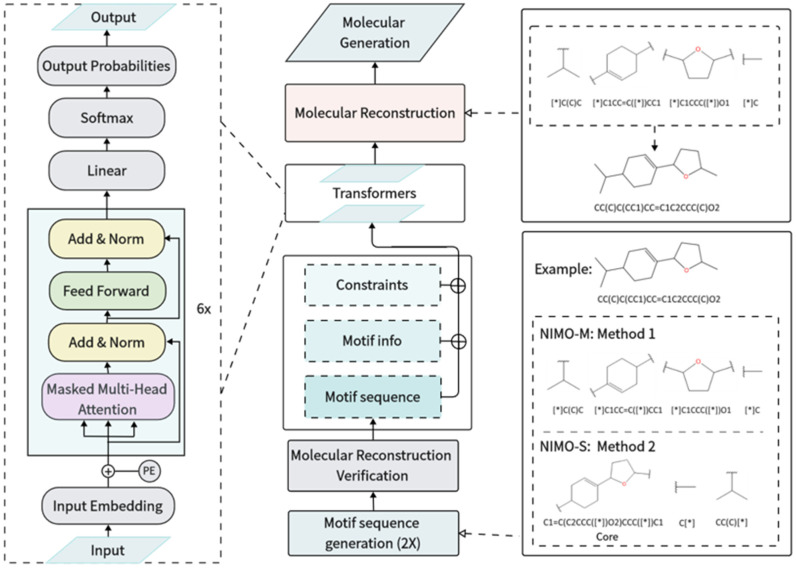
The workflow of NIMO. Middle: The model-training preparation process includes motif sequence canonicalization and molecular reconstruction verification, which results in unique canonical motif sequences. Then, each input sequence includes constraints, motif information, and a motif sequence, which are used to train a conditional transformer architecture. During the model sampling phase, the sampled motifs undergo molecular reconstruction to form the final complete molecules. Left: The architecture of the conditional transformer neural network. Right: the examples of molecular reconstruction and motif sequence generation (including two methods, corresponding to NIMO-M and NIMO-S). “*” denotes a dummy atom within the motif.

**Table 1 molecules-29-01867-t001:** The conditional and MOSES evaluation metrics for the generated molecules. The detailed definitions are provided in Appendix A.

	Models	NIMO-M	NIMO-S	MCMG	QBMG	FBMG
Conditional metrics	Validity	94.5%	**99.3%**	95.0%	94.5%	42.9%
	Uniqueness	99.7%	99.1%	98.4%	**99.9%**	98.5%
	Novelty	61.0%	**77.8%**	65.7%	42.2%	99.9%
MOSES metrics	FCD↓ ^a^	**3.71**	11.2	4.52	19.2	6.11
	SNN↓	0.87	**0.65**	0.71	0.95	0.51
	Frag↓	0.85	**0.77**	0.95	0.99	0.48
	Scaf↓	0.67	0.83	**0.65**	0.66	0.57
	IntDiv	**88.3%**	86.5%	87.8%	86.6%	73.9%
	Novelty	71.4%	**89.0%**	79.5%	52.4%	99.9%
	SAS↓	**0.78**	0.91	1.22	0.87	0.94

^a^ ↓ The lower, the better. FCD refers to “Fréchet ChemNet Distance”, which is a metric to predict biological activities based on a deep neural network; SNN refers to “nearest neighbor similarity”; Frag/Scaf refers to “fragment/scaffold similarity”; IntDiv refers to “internal diversity”. Bold text indicates the best result.

**Table 2 molecules-29-01867-t002:** The performance for terpenoid generation. “Success” means the success rate of molecules predicted as terpenoids by NPClassifier; “coverage” represents the proportion of the number of unique RSs/FGs extracted from the generated set and existing in the training set to the total RSs/FGs of the generated set; “recovery” represents the proportion of the number of unique RSs/FGs extracted from the generated set and existing in the training set to the unique RSs/FGs of the generated set.

	Metrics	NIMO-S	NIMO-S’	MCMG	QBMG
Terpenoids	Success	91.9%	**95.4%**	71.2%	89.7%
Ring systems (RSs) ^a^	Coverage	27.5%	**29.8%**	28.1%	8.3%
	Recovery	**99.4%**	69.5%	62.4%	10.6%
Functional groups (FGs) ^b^	Coverage	5.9%	**6.2%**	4.3%	4.9%
	Recovery	**93.2%**	89.7%	58.1%	47.1%

^a,b^ RSs and FGs were automatically extracted based on RDKit in an unbiased way. Bold text indicates the best result.

**Table 3 molecules-29-01867-t003:** Summary of anti-malarial activity-oriented molecular generation.

	Train	NIMO-M	MCMG	NIMO-M’
Samples	744,986	5000	5000	1000
EF [50%] ^a^	20.07	46.82	44.99	68.22
EF [10%]	44.36	72.09	69.11	81.33
EF [1%]	80.4	81.97	89.17	92.21
Active %	10.0%	55.9%	52.1%	85.5%

^a^ EF means enrichment factor provided by MAIP. EF [X] is the hit rate (the proportion of active compounds) within a defined sorted fraction divided by the total hit rate.

**Table 4 molecules-29-01867-t004:** MOE docking result of compounds generated by NIMO-M.

PDB	2HMG (CHEMBL2902)		1BO7 (CHEMBL5328)	
Compounds	1000	5000	1000	5000
Predicted candidates	15	82	93	294
Docking score < native	10	26	10	23
RMSD < 2	10	65	48	104

## Data Availability

The code of NIMO is publicly available from the GitHub repository: https://github.com/shenxj9/NIMO (accessed on 22 March 2024).

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
