# Peer review of "NIMO: A Natural Product-Inspired Molecular Generative Model Based on Conditional Transformer"

_molecules, 2024, doi:10.3390/molecules29081867_

Round 1
Reviewer 1 Report
Comments and Suggestions for Authors Drawing inspiration from the concept of fragments-assembled pseudo-natural products, Shen et al. developed a computational tool named NIMO, leveraging transformer neural network models. NIMO utilizes two specialized motif extraction methods to translate molecular graphs into semantic motif sequences. These sequences are then employed to train molecular generative models, enabling recognition of syntactic patterns and structure-property relationships. Across various task scenarios, NIMO models were trained and evaluated for tasks such as structure-guided, activity-oriented, and pocket-based molecule generation. Our results demonstrate NIMO's outstanding performance in generating molecules from scratch and optimizing structures from scaffolds. Additionally, the paper presents impressive figures. Here are my comments:1.Ensure that the latest papers are cited in the Introduction. I noticed that the articles cited by the author were all published before 2020.
2.Consider the necessity of Figure 5a. If comparing Figure 5a and 5b can provide new insights, it might be worth including. However, if it primarily involves data processing, it could be placed in the appendix.
3.In Figure 1, the author claims to have used deep learning, but lacks specific details on the method. Also, should provide more information on the reconstruction method used, allowing others to repeat the experiments. Although the author has verified the results, it's essential to include these details for reproducibility.
4.The discussion section should compare data with other peer review papers and perform verification while explaining the novelty of the your findings.
5.The author should provide a specific description of the method used for fragmentation in the experimental section.
6. The author can give some prospects or provide potential applications of the conclusions.
7.Make slight modifications to the English language for clarity and coherence.
Reviewer 2 Report
Comments and Suggestions for Authors
This manuscript is describing about NIMO to expend the diversity of the natural compouds, inspiring by the concept of fragments-assembled pseudo-natural products. Using the NIMO, motifs were extracted and used for molecular generative models. The authors explored the performance of NIMO in structure-guided, activity-oriented, and pocket-based molecule generation tasks. It was written well and it is recommended to be acceptable. To improve the paper, leading compoud(s) can be tested for biological activities, not just in silico prediction.
